# Neural Networks with Block Diagonal Inner Product Layers

## Abstract

Artificial neural networks have opened up a world of possibilities in data science and artificial intelligence, but neural networks are cumbersome tools that grow with the complexity of the learning problem. We make contributions to this issue by considering a modified version of the fully connected layer we call a block diagonal inner product layer. These modified layers have weight matrices that are block diagonal, turning a single fully connected layer into a set of densely connected neuron groups. This idea is a natural extension of group, or depthwise separable, convolutional layers applied to the fully connected layers. Block diagonal inner product layers can be achieved by either initializing a purely block diagonal weight matrix or by iteratively pruning off diagonal block entries. This method condenses network storage and speeds up the run time without significant adverse effect on the testing accuracy, thus offering a new approach to improve network computation efficiency.

## 1 Introduction

Today, it is well known that larger neural networks can better represent complex data and hence achieve higher accuracy than smaller networks (Hornik et al., 1989; Simonyan & Zisserman, 2014; Sermanet et al., 2014). While larger networks are more capable than their smaller counterparts, their size consumes significant storage and computational resources and memory bandwidth. Ideally, efforts to reduce memory requirements would also lessen computational demand, but often these competing interests force a trade-off. The fully connected layers are unwieldy, yet they continue to be present in the most successful networks (Krizhevsky et al., 2012; Zeiler & Fergus, 2013; Simonyan & Zisserman, 2014). Our work addresses both memory and computational demand without compromise. Focusing our attention on the inner product layers, we decrease network memory footprint and improve network computational demand.

While larger network architectures achieve higher accuracy, there are a variety of methods to condense them without much harm to the network accuracy. One such technique that has gained popularity is pruning (Reed, 1993; Han et al., 2015a;b), but traditional pruning has disadvantages related to network runtime. Most existing pruning processes significantly slow down network training, and the final trained network is usually slower to execute (Han et al., 2015a). Sparse format operations require additional overhead that can greatly slow down performance unless one prunes nearly all weight entries, which can damage network accuracy.

Localized memory access patterns can be computed faster than non-localized lookups. By implementing block diagonal inner product layers in place of fully connected layers, we condense neural networks in a structured manner that speeds up the final runtime and does little harm to the final accuracy. Block diagonal inner product layers can be implemented by either initializing a purely block diagonal weight matrix or by initializing a fully connected layer and focusing pruning efforts off the diagonal blocks to coax the dense weight matrix into structured sparsity. The first method also reduces the gradient computation time and hence the overall training time. The latter method can improve accuracy and supports the robustness of networks to *shaping*. That is, pruning can be used as a mapping between architectures–in particular, a mapping to more convenient architectures. Depending on how many iterations the pruning process takes, this method may also speed up training. We have converted a single fully connected layer into a group of smaller inner product learners whose combined efforts form a stronger learner, in essence boosting the layer. These methods also

bring artificial neural networks closer to the architecture of biological mammalian brains, which have more local connectivity (Herculano-Houzel, 2012).

## 2 RELATED WORK

Weight pruning comes in many forms including penalty and second derivative methods, sensitivity analysis and cross validation; it can be done iteratively during training or to a trained network before refining the surviving weights (Reed, 1993; Castellano et al., 1997; LeCun et al., 1990; Hassibi & Stork, 1992; Engelbrecht, 2001). With any of these methods, the result is a sparse network that takes less storage space than its fully connected counterpart. Han et al. (2015b) iteratively prune a network using the penalty method by adding a mask that disregards pruned parameters for each weight tensor. This means that the number of required floating point operations decreases, but the number performed stays the same. Furthermore, masking out updates takes additional time. Quantization and Huffman coding can be used to compress a trained network further (Xie & Jabri, 1992; Han et al., 2015a). Han et al. (2015a) report the average time spent on a forward propagation after pruning is complete and the resulting sparse layers have been converted to CSR format; for batch sizes larger than one, the sparse computations are significantly slower than the dense calculations.

Node pruning (He et al., 2014; Srinivas & Babu, 2015) could be used to speed up training and final execution time more easily than weight pruning since node pruning preserves some structure, but drastic node pruning can harm the network accuracy. In practice, node pruning requires additional weight fine-tuning to maintain accuracy. Other approaches include storing a low rank approximation for a layer's weight matrix (Sainath et al., 2013), using specific parameter sharing mechanisms (Chen et al., 2015), training smaller models on outputs of larger models (distillation) (Hinton et al., 2014), allowing neurons to read stale gradient updates (Ho et al., 2013) and using lower precision (Vanhoucke et al., 2011; M.Courbariaux et al., 2015; Gupta et al., 2015). Note that using lower precision or stale gradients are techniques that can be applied to any of the methods considered, including ours.

More recently, there has been momentum in the direction of structured reduction of network architecture. Sindhwani et al. (2015) propose structured parameter matrices characterized by low displacement rank that yield high compression rate as well as fast forward and gradient evaluation. Their work focuses on toeplitz-related transforms of the fully connected layer weight matrix. However, speedup is generally only seen for compression of large weight matrices, and their structured sparsity methods require parameter sharing which may be undesirable in some cases. Block sparsity yields compression with parameter flexibility. Group lasso has made contributions to the compression of convolutional layers (Yuan & Lin, 2006; Lebedev & Lempitsky, 2016; Wen et al., 2016). Group lasso expands the concept of node pruning to convolutional filters. That is, group lasso applies $L_1$ norm regularization to entire filters. Likewise, group, or depthwise separable, convolutions have been used in recent CNN architectures with great success (Xie et al., 2016; Zhang et al., 2017; Chollet, 2017; Howard et al., 2017). In group convolutions, the number of nonzero weights does not change, but the number of connections changes; a particular filter does not see all of the channels of the previous layer. Block diagonal inner product layers apply this idea of separable neuron groups to the fully connected layers, which decreases the number of weights as well as the number of connections. Decoupling channels in images is natural because channels often carry very similar information. When converting a fully connected layer to a block diagonal inner product layer, blocks may not even see a whole channel. This method transforms a fully connected layer into an ensemble of smaller fully connected neuron groups that boost the layer. Methods to condense and speedup convolutional layers like group lasso or group convolutions, can also be used in conjunction with our methods.

## 3 METHODOLOGY

We consider two methods for implementing block diagonal inner product layers:

1. We initialize a layer with a purely block diagonal weight matrix and keep the number of connections constant throughout training.

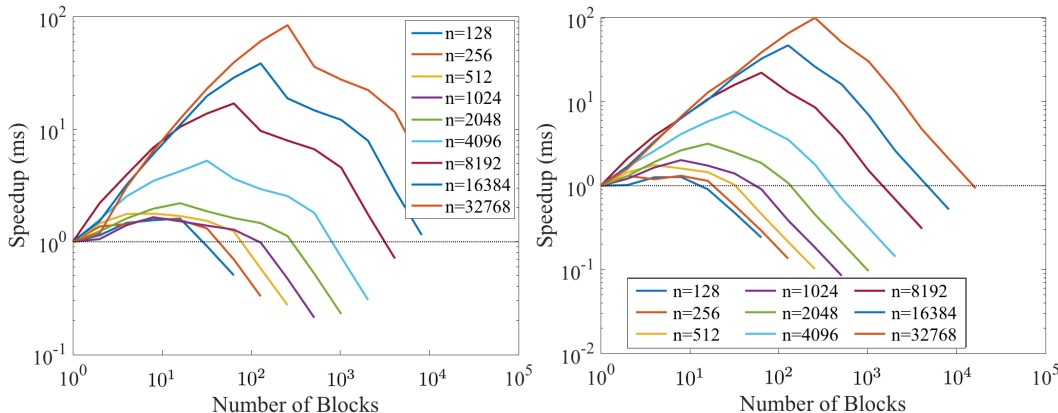

Figure 1: Speedup when performing matrix multiplication using an $n \times n$ weight matrix and batch size 100. (Left) Speedup when performing only one forward matrix product. (Right) Speedup when performing all three matrix products involved in the forward and backward pass in gradient descent.

2. We initialize a fully connected layer and iteratively prune entries off the diagonal blocks to achieve a block substructure.

Within a layer, all blocks have the same size. Method 2 is accomplished in three phases: a dense phase, an iterative pruning phase and a block diagonal phase. In the dense phase a fully connected layer is initialized and trained in the standard way. During the iterative pruning phase, focused pruning is applied to entries off the diagonal blocks using the weight decay method with $L_1$-norm. That is, if $W$ is the weight matrix for a fully connected layer we wish to push toward block diagonal, we add

$$\alpha \sum_{i,j} |\mathbb{1}_{i,j} W_{i,j}| \tag{1}$$

to the loss function during the iterative pruning phase, where $\alpha$ is a tuning parameter and $\mathbb{1}_{i,j}$ indicates whether $W_{i,j}$ is off the diagonal blocks in $W$. The frequency of regularization and pruning during this phase are additional hyperparameters. During this phase, masking out updates for pruned entries is more efficient than maintaining sparse format. When pruning is complete, to maximize speedup it is best to reformat the weight matrix once such that the blocks are condensed and adjacent in memory.[1] Batched smaller dense calculations for the blocks use cuBLAS strided batched multiplication (Nickolls et al., 2008). There is a lot of flexibility in method 2 that can be tuned for specific user needs. More pruning iterations may increase the total training time but can yield higher accuracy and reduce overfitting.

## 4 EXPERIMENTS: SPEEDUP AND ACCURACY

Our goal is to reduce memory storage of the inner product layers while maintaining or reducing the final execution time of the network with minimal loss in accuracy. We will also see the reduction of total training time in some cases. To test this, we ran experiments on the MNIST (LeCun et al.), CIFAR10 (Krizhevsky, 2009) and ImageNet (Russakovsky et al., 2015) datasets. All experiments are run on the Bridges' NVIDIA P100 GPUs through the Pittsburgh Supercomputing Center. We implement our work in Caffe (Jia et al., 2014) which provides suggested network architectures for these datasets. Training is done with batched gradient descent using the cross-entropy loss function on the softmax of the output layer. We report the forward time per inner product layer when the layers are purely block diagonal; this time only measures the matrix multiplication in the forward step and does not include the time to prune. We compare these times to the runtime of sparse matrix multiplication with random entries in CSR format using cuSPARSE (Nickolls et al., 2008). We also report the combined forward and backward time to do the three matrix products involved in gradient

---

[1] When using block diagonal layers, one should alter the output format of the previous layer and the expected input format of the following layer accordingly, in particular to row major ordering.

descent training when the layers are purely block diagonal. For brevity we refer to the block diagonal method as $(b_1, \ldots, b_n)$-BD; $b_i = 1$ indicates that layer $i$ is fully connected. FC is short for all inner product layers being fully connected. We refer to networks that differ only by the number of blocks in a single inner product layer as *sister* networks.

Figure 1 shows the speedup when performing matrix multiplication using an $n \times n$ weight matrix and batch size 100 when the weight matrix is purely block diagonal. The speedup when performing only the forward-pass matrix product is shown in the left pane, and the speedup when performing all gradient descent products is shown in the right pane. As the number of blocks increases, the overhead to perform cuBLAS strided batched multiplication can become noticeable; this library is not yet well optimized for performing many small matrix products (Masliah et al., 2016). However, with specialized batched multiplications for many small matrices, Jhurani & Mullowney (2015) attain up to 6 fold speedup.

Zhang et al. (2017) note that when multiple group convolutions are stacked together, outputs from a particular group only see the inputs within the group. They suggest that this property blocks information flow between channel groups and weakens representation. To correct for this, they suggest dividing the channels in each group into subgroups, and feeding each group in the next layer with a mixture of subgroups from various parent groups in the previous layer. One drawback of applying this approach to block inner product layers is that it either requires moving entries in memory or doing more, smaller matrix products. As mentioned, performing many small matrix products can hurt performance; this can be seen in Figure 1. Using pruning to achieve the block diagonal structure, as in method 2, also addresses this issue. Pruning does add some work to the training iterations, but does not add work to the final execution of the trained network unlike the shuffling method found in ShuffleNet (Zhang et al., 2017). After pruning is complete, the learned weights are the result of a more complete picture; while the information flow has been constrained, it is preserved like a ghost in the remaining weights. Another alternative to the fixed channel shuffle offered in ShuffleNet, is a random shuffle of the whole blocks similar to the "random sparse convolution" layer in the CNN library *cuda-convnet* (Krizhevsky, 2012a). We will compare results using all of these methods.

## 4.1 MNIST

We experimented on the MNIST dataset with the LeNet-5 framework (LeCun et al., 1998). LeNet-5 has two convolutional layers with pooling followed by two inner product layers with ReLU activation. The first inner product layer, ip1, has a $500 \times 800$ weight matrix, and the output inner product layer, ip2, has a $10 \times 500$ weight matrix. We initialize the inner product layers using the Xavier weight filler (Glorot & Bengio, 2010), which samples a uniform distribution with variance $1/n_{in}$, where $n_{in}$ is the number of neurons feeding into a node. We used a training batch size of 64 and all other hyperparameters provided by Caffe are left unchanged.

Figure 2 shows time and accuracy results for block diagonal method 1 without pruning. The $(b_1, b_2)$-BD architecture has $(800 \times 500)/b_1 + (500 \times 10)/b_2$ nonzero weights across both inner product layers. The points at which the forward sparse and forward block curves meet in Figure 2 (left) indicate the fully connected dense forward runtimes for each layer; these are made clearer with dotted, black, vertical lines. There is $\geq 1.4\times$ speedup for $b_1 \leq 50$, or 8000 nonzero entries, when timing both forward and backward matrix products, and $1.6\times$ speedup when $b_1 = 100$, or 4000 nonzero entries, in the forward only case. Sparse format times are slow until there are less than 50 nonzero entries. Figure 2 (right) shows a slight decline in accuracy as the number of nonzero entries decreases. FC achieves a final accuracy of 99.11%. Without pruning, $(100, 10)$-BD has a final accuracy of 98.52%. In all cases testing accuracy remains within 1% of FC accuracy.

Using traditional iterative pruning with $L_2$ regularization, as suggested in (Han et al., 2015b), pruning every fifth iteration until 4000 and 500 nonzero entries survived in ip1 and ip2 respectively gave an accuracy of 98.55%, but the forward multiplication was more than 8 times slower than the dense fully connected case. Implementing $(100, 10)$-BD method 2 with pruning using 15 dense iterations and 350 pruning iterations gave a final accuracy of 98.65%. Thus we saw no great benefit to using pruning over initializing pure block diagonal inner product layers for the MNIST dataset. In this setting, using neither random shuffling of whole blocks in ip1, nor fixed sub-block shuffling in the manner of Zhang et al. (2017) delivered any noticeable improvement.

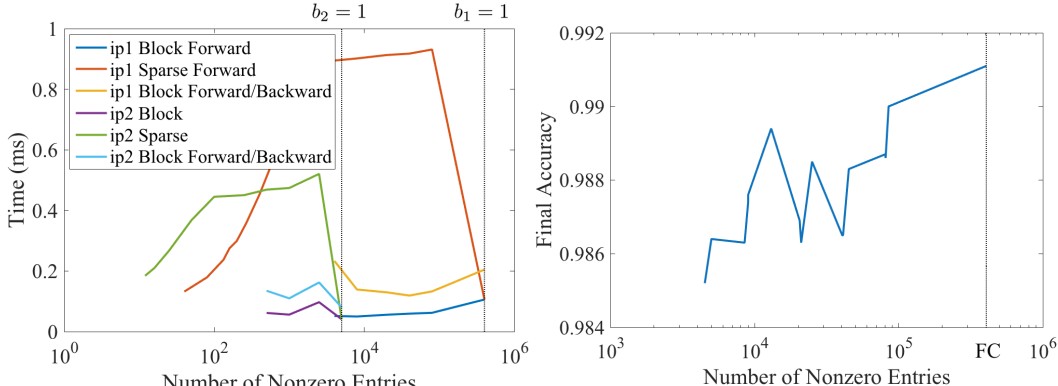

Figure 2: Time/Accuracy results using Lenet-5 on MNIST with batch size 64. (Left) For each inner product layer: forward runtimes of block diagonal and CSR sparse formats, combined forward and backward runtimes of block diagonal format. (Right) Accuracy versus total number of nonzero entries in the inner product layers after 10000 training iterations using block diagonal method 1.

In (Sindhwani et al., 2015), Toeplitz (3) has error rate $2.09\%$ using a single hidden layer net with 1000 hidden nodes on MNIST. This method yields 63.32 fold compression over the fully connected setting. However from their Figure 3, this slows down the forward pass by around $1.5\times$ and the backward pass by around $5.5\times$. Our net with one hidden layer, 980 hidden nodes using $(49, 1)$-BD on MNIST has 29.43 fold compression and error rate $4.37\%$ using method 2 with pruning. This is striking because the blocks of neurons in the hidden layer can only see a portion of the test input images. Our speedup is 1.53 for forward only and 1.04 when combining the forward and backward runtime.

As mentioned we left suggested hyperpameters by Caffe unchanged for hyperparameters like learning rate, momentum and weight decay. In our experiments on the MNIST dataset we performed only manual hyperparameter tuning of new hyperparameters introduced by method 2 like the coefficient of the new regularization term (see equation 1) and the frequency of pruning iterations.

## 4.2 CIFAR10

We experimented on the CIFAR10 dataset with Krizhevsky's cuda-convnet (2012b). Cuda-convnet has three convolutional layers with ReLu activation and pooling, followed by two fully connected layers with no activation. The first inner product layer, ip1, has a $64 \times 1024$ weight matrix, and the second has a $10 \times 64$ weight matrix. In all methods the inner product layer weights are initialized using a Gaussian filler with standard deviation 0.1, as suggested by Caffe. We used a training batch size of 100, and all other hyperparameters provided by Caffe's "quick" model are left unchanged. Caffe reports $\approx 75\%$ accuracy with the FC architecture.

Figure 3 shows time and accuracy results for the block diagonal method 1 without pruning. The $(b_1, b_2)$-BD architecture has $(1024 \times 64)/b_1 + (64 \times 10)/b_2$ nonzero entries across both inner product layers. In the ip1 layer, there is $\geq 1.26\times$ speedup for $b_1 \leq 32$, or 2048 nonzero entries, when timing both forward and backward matrix products, and $\geq 1.65\times$ speedup for $b_1 \leq 64$, or 1024 nonzero entries, in the forward only case. Again, sparse format performs poorly until there are less than 50 nonzero entries. Figure 3 (right) shows a decline in accuracy as the blocks increase. FC achieves a final accuracy of $76.29\%$. Without pruning, $(64, 2)$-BD has a final accuracy of $72.49\%$.

Using traditional iterative pruning with $L_2$ regularization pruning every fifth iteration until 1024 and 320 nonzero entries survived in the final two inner product layers gave an accuracy of $75.18\%$, but again the forward multiplication was more than 8 times slower than the dense fully connected computation. On the other hand, implementing $(64, 2)$-BD method 2 with pruning, which has corresponding numbers of nonzero entries, with 500 dense iterations and 1000 pruning iterations gave a final accuracy of $74.81\%$. This is a 35.97 fold compression of the inner product layer parameters with only a $1.5\%$ drop in accuracy. The total forward runtime of ip1 and ip2 in $(64, 2)$-BD is 1.6 times faster than in FC. To achieve comparable speed with sparse format we used traditional itera-

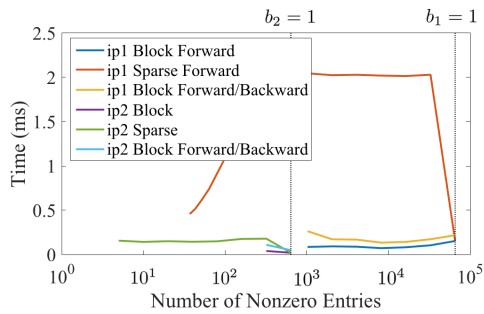 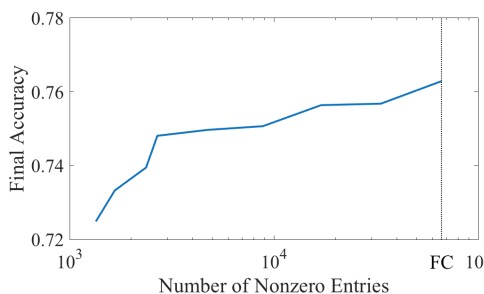

Figure 3: Time/Accuracy results of cuda-convnet on CIFAR10 with batch size 100. (Left) For each inner product layer: forward runtimes of block diagonal and CSR sparse formats, combined forward and backward runtimes of block diagonal format. (Right) Accuracy versus total number of nonzero entries in the inner product layers after 9000 training iterations using block diagonal method 1.

tive pruning to leave 37 and 40 nonzero entries in the final inner product layers giving an accuracy of 73.01%. Thus implementing block diagonal layers with pruning yields comparable accuracy and memory condensation to traditional iterative pruning with faster final execution time.

Whole node pruning decreases the accuracy more than corresponding reductions in the block diagonal setting. Node pruning until ip1 had only 2 outputs, i.e. a $1024 \times 2$ weight matrix, and ip2 had a $2 \times 10$ weight matrix for a total of 2068 weights between the two layers gave a final accuracy of 59.67%. (64,2)-BD has a total of 1344 weights between the two inner product layers and had a final accuracy 15.14% higher with pruning.

The final accuracy on an independent test set was 76.29% on CIFAR10 using the FC net while the final accuracy on the training set itself was 83.32%. Using the (64,2)-BD net without pruning, the accuracy on an independent test set was 72.49%, but on the training set was 75.63%. With pruning, the accuracy of (64,2)-BD on an independent test set was 74.81%, but on the training set was 76.85%. Both block diagonal methods decrease overfitting; the block diagonal method with pruning decreases overfitting slightly more.

We saw no significant affect on the final accuracy after 9000 iterations when using random shuffling of the ip2 layer blocks during training. Similarly, performing fixed sub-block shuffle as suggested by Zhang et al. (2017) on the $(2, 2)$-BD architecture using method 1 where shuffling is applied to sub-blocks of layer ip2 did not yield improvement over $(2, 2)$-BD without fixed sub-block shuffling. These methods may be useful in networks with many stacked block inner product layers or networks with sparse convolution as a result of group lasso or group convolution, but saw no benefit in the cuda-convnet architecture with two inner product layers. Using method 2 to achieve a block substructure with pruning did yield improvement over method 1.

### 4.3 IMAGENET : INCOMPLETE

In this section we will put accuracy results when we are able to get them. Currently we are unable to run ImageNet experiments using AlexNet because we need more memory on the Bridges Supercomputer. We have placed a request for more memory and we are waiting for that request to be granted.

We experimented on the ImageNet dataset with the AlexNet framework (Krizhevsky et al., 2012). AlexNet has five convolutional layers followed by three inner product layers with ReLU and dropout activation. The first inner product layer, ip1, has a $4096 \times 9216$ weight matrix, the second inner product layer, ip2, has a $4096 \times 4096$ weight matrix, and the output inner product layer, ip3, has a $1000 \times 4096$ weight matrix. We initialize the inner product layers using a gaussian weight filler. We used a training batch size of 256 and all other hyperparameters provided by Caffe are left unchanged.

Figure 4 shows time and accuracy results for block diagonal method 1 without pruning. The $(b_1, b_2, b_3)$-BD architecture has $(4096 \times 9216)/b_1 + (4096 \times 4096)/b_2 + (1000 \times 4096)/b_3$ nonzero weights across all inner product layers. The points at which the forward sparse and forward block curves meet in Figure 4 (left) indicate the fully connected dense forward runtimes for each layer.

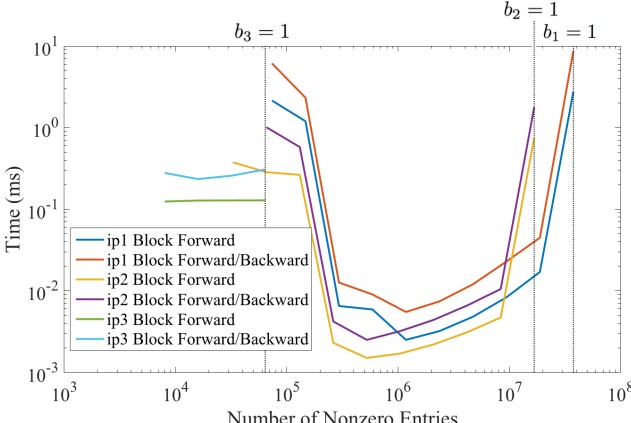

Figure 4: Time/Accuracy results using alexnet on imagenet with batch size 256. (Left) For each inner product layer: forward runtimes as well as combined forward and backward runtimes of block diagonal format. (Right) Accuracy versus total number of nonzero entries in the inner product layers after 360000 training iterations using block diagonal method 1.

## 5  CONCLUSION

We have shown that block diagonal inner product layers can reduce network size, training time and final execution time without significant harm to the network performance.

While traditional iterative pruning can reduce storage, the random indices of the surviving weights make sparse computation inefficient, slowing down the training and final execution time of the network. Our block diagonal methods address this inefficiency by confining dense regions to blocks along the diagonal. Without pruning, block diagonal method 1 allows for faster training time. Method 2 preserves the learning with focused, structured pruning that reduces computation for speedup during execution. In our experiments, method 2 saw higher accuracy than the purely block diagonal method for the more complex learning problem, CIFAR10; however, the increase in accuracy came in exchange for slightly more time to train the network. There is great deal of flexibility in our block diagonal methods that can be tuned to an individual problem. These methods may also make larger network architectures more feasible to train and use since they convert a fully connected layer into a collection of smaller inner product learners working jointly to form a stronger learner. In particular, GPU memory constraints become less constricting.

There is a lot of room for additional speedup with block diagonal layers. Dependency between layers poses a noteworthy bottleneck in network parallelization. With structured sparsity like ours, one no longer needs a full barrier between layers. Additional speedup would be seen in software optimized to support weight matrices with organized sparse form, such as blocks, rather than being optimized for dense matrices. For example, for many small blocks, one can reach up to 6 fold speedup with specialized batched matrix multiplication (Jhurani & Mullowney, 2015). Hardware has been developing to better support sparse operations. Block format may be especially suitable for training on evolving architectures such as neuromorphic systems. These systems, which are far more efficient than GPUs at simulating mammalian brains, have a pronounced 2-D structure and are ill-suited to large dense matrix calculations (Merolla et al., 2014; Boahen, 2014).

ACKNOWLEDGMENTS

All experiments are run on the Bridges' NVIDIA P100 GPUs through the Pittsburgh Supercomputing Center.

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
