# OpenReview forum: "Neural Networks with Block Diagonal Inner Product Layers"
_ICLR.cc/2018/Conference — Reject_

### Official Review · AnonReviewer3 · 2017-11-26
**A mostly experimental paper using block diagonal weight matrices for NN pruning, with valuable insights into random matrix theory to model weight behavior.**

**Rating:** 5
**Confidence:** 4

**Review:**

This is a mostly experimental paper which evaluates the capabilities of neural networks with weight matrices that are block diagonal. The authors describe two methods to obtain this structure: (1) enforced during training, (2) enforced through regularization and pruning. As a second contribution, the authors show experimentally that the random matrix theory can provide a good model of the spectral behavior of the weight matrix when it is large. However, the authors only conjecture as to the potential of this method without describing clear ways of approaching this subject, which somewhat lessens the strength of their argument.

Quality: this paper is of good quality
Clarity: this paper is clear, but would benefit from better figures and from tables to report the numerical results instead of inserting them into plain text.
Originality: this paper introduces block diagonal matrices to structure the weights of a neural network. The idea of structured matrices in this context is not new, but the diagonal block structure appears to be.
Significance: This paper is somewhat significant.

PROS
- A new approach to analyzing the behavior of weight matrices during learning
- A new structure for weight matrices that provides good performance while reducing matrix storage requirements and speeding up forward and backward passes.

CONS
- Some of the figures are hard to read (in particular Fig 1 & 2 left) and would benefit from a better layout.
- It would be valuable to see experiments on bigger datasets than only MNIST and CIFAR-10.
- I understand that the main advantage of this method is the speedup; however, providing the final accuracy as a function of the nonzero entries for slower methods (e.g. the sparse pruning showed in Fig 1. a) would provide a more complete picture.

Main questions:
- Could you briefly comment on the training time in section 4.1?
- Could you elaborate on the last sentence of section 4.1?
- You state: "singular values of an IP layer behave according to the MP distribution even after 1000s of training iterations." Is this a known fact, or something that you observed empirically? In practice, how large must the weight matrix be to observe this behavior?

Nitpicks:
- I believe the term "fully connected" is more standard than "inner product" and would add clarity to the paper, but I may be mistaken.

---

> ### Author Response · Authors · 2017-12-21
> **Split the paper into two separate papers**
>
> Thank you for your thorough review.
>
> At another reviewer’s suggestion, we have chose to split the random matrix theory and the block diagonal inner product layer work into two separate papers.  We have decided to submit only the block diagonal inner product layer work for this review.
>
> Re: “Some of the figures are hard to read (in particular Fig 1 & 2 left) and would benefit from a better layout.”
>
> We improved these figures by moving the labels indicating the values when the layer is fully connected (i.e. blocks=1).
>
> Re: It would be valuable to see experiments on bigger datasets than only MNIST and CIFAR-10.
>
> We plan to run experiments on the ImageNet dataset using AlexNet to demonstrate our work in a ‘large’ setting. However, we are currently waiting on a larger memory allocation on the Bridges supercomputer to handle this task. This allocation is granted, but will take up to 5 days to be active. We will update the paper with these results when we have them.
>
> Re: I understand that the main advantage of this method is the speedup; however, providing the final accuracy as a function of the nonzero entries for slower methods (e.g. the sparse pruning showed in Fig 1. a) would provide a more complete picture.
>
> We offered two data points here for each dataset: the accuracy when pruning that yeilds comparable speed and the accuracy for comparable pruning.  Comparable pruning (for random entries) is consistenly more accurate, but can have 8x slower execution time.
>
>
> Re: Could you briefly comment on the training time in section 4.1?
>
> We added an additional figure that may be helpful here.  Figure 1 shows how block diagonal inner product layers scale for large weight matrices matrices. We discuss Figure 1 in Section 4: Experiments, Paragraph 2
>
> Re: Could you elaborate on the last sentence of section 4.1?
>
> Here we just wanted to note that an error rate of 4.37% is impressive considering the serious contrain on the flow of information when the first layer is unable to get a full picture.
>
> Re: “I believe the term "fully connected" is more standard than "inner product" and would add clarity to the paper, but I may be mistaken.”
>
> The term fully connected is more standard, but this name is descriptive.  A layer is fully connected if every node in that layer has a weight for every node in the previous layer, since this is not the case for our block inner product layer we thought using the term, “fully connected” would be misleading.  We do offer this term in the beginning of the paper for context.
>
>
>
>
>
> We have uploaded a revised (working) version of the paper focusing on the block diagonal inner product layer results.
>
> We list the major changes here:
>
> Section 2: Related Work, Paragraph 3
> Here we focus on the related work we deem most similar to our own. We discuss Group Lasso, Group Convolution and Sindhwani et al. (2015) work with Toeplitz-like transforms as successful options to reduce network size in a structured manner.
>
> Section 4: Experiments, Paragraph 2
> We discuss the Figure 1.  This new figure shows how block diagonal inner product layers scale for large weight matrices matrices.
>
> Section 4: Experiments, Paragraph 3
> Here we examine methods to improve the flow of information in sprase architecture and compare them to block diagonal inner product layer method 2 with pruning. These inclue fixed sub-block shuffling inspired by channel shuffling in Zhang et al. (2017) and random block shuffling.
>
> 4.2 CIFAR10, Paragraph 6
> Here we show that block diagonal inner product layer method 2 with pruning shows improved results over fixed sub-block shuffling inspired by channel shuffling in Zhang et al. (2017) and random block shuffling, which are other attempts to improve the flow of information in sparse architecture as discussed in Section 4: Experiments, Paragraph 3.

---

### Official Review · AnonReviewer2 · 2017-11-27
**Block diagonal is more efficient that sparse formats and can be used effectively; other parts of paper are vague**

**Rating:** 6
**Confidence:** 4

**Review:**

The paper proposes to make the inner layers in a neural network be block diagonal, mainly as an alternative to pruning. The implementation of this seems straightforward, and can be done either via initialization or via pruning on the off-diagonals. There are a few ideas the paper discusses:

(1) compared to pruning weight matrices and making them sparse, block diagonal matrices are more efficient since they utilize level 3 BLAS rather than sparse operations which have significant overhead and are not "worth it" until the matrix is extremely sparse. I think this case is well supported via their experiments, and I largely agree.

(2) that therefore, block diagonal layers lead to more efficient networks. This point is murkier, because the paper doesn't discuss possible increases in *training time* (due to increased number of iterations) in much detail. At if we only care about running the net, then reducing the time from 0.4s to 0.2s doesn't seem to be that useful (maybe it is for real-time predictions? Please cite some work in that case)

(3) to summarize points (1) and (2), block diagonal architectures are a nice alternative to pruned architectures, with similar accuracy, and more benefit to speed (mainly speed at run-time, or speed of a single iteration, not necessarily speed to train)

[as I am not primarly a neural net researcher, I had always thought pruning was done to decrease over-fitting, not to increase computation speed, so this was a surprise to me; also note that the sparse matrix format can increase runtime if implemented as a sparse object, as demonstrated in this paper, but one could always pretend it is sparse, so you never ought to be slower with a sparse matrix]

(4) there is some vague connection to random matrices, with some limited experiments that are consistent with this observation but far from establish it, and without any theoretical analysis (Martingale or Markov chain theory)

This is an experimental/methods paper that proposes a new algorithm, explained only in general details, and backs up it up with two reasonable experiments (that do a good job of convincing me of point (1) above). The authors seem to restrict themselves to convolutional networks in the first paragraph (and experiments) but don't discuss the implications or reasons of this assumption. The authors seem to understand the literature well, and not being an expert myself, I have the impression they are doing a fair job.


The paper could have gone farther experimentally (or theoretically) in my opinion. For example, with sparse and block diagonal matrices, reducing the size of the matrix to fit into the cache on the GPU must obviously make a difference, but this did not seem to be investigated. I was also wondering about when 2 or more layers are block sparse, do these blocks overlap? i.e., are they randomly permuted between layers so that the blocks mix? And even with a single block, does it matter what permutation you use? (or perhaps does it not matter due to the convolutional structure?)

The section on the variance of the weights is rather unclear mathematically, starting with the abstract and even continuing into the paper. We are talking about sample variance? What does DeltaVar mean in eq (2)? The Marchenko-Pastur theorem seemed to even be imprecise, since if y>1, then a < 0, implying that there is a nonzero chance that the positive semi-definite matrix XX' has a negative eigenvalue.

I agree this relationship with random matrices could be interesting, but it seems too vague right now. Is there some central limit theorem explanation? Are you sure that you've run enough iterations to fully converge? (Fig 4 was still trending up for b1=64). Was it due to the convolutional net structure (you could test this)? Or, perhaps train a network on two datasets, one which is not learnable (iid random labels), and one which is very easily learnable (e.g., linearly separable). Would this affect the distributions?

Furthermore, I think I misunderstood parts, because the scaling in MNIST and CIFAR was different and I didn't see why (for MNIST, it was proportional to block size, and for CIFAR it was independent of block size almost).

Minor comment: last paragraph of 4.1, comparing with Sindhwani et al., was confusing to me. Why was this mentioned? And it doesn't seem to be comparable. I have no idea what "Toeplitz (3)" is.

---

> ### Author Response · Authors · 2017-12-20
> **Split the paper into two separate papers**
>
> Thank you for your thorough review.
>
> At another reviewer’s suggestion we have chose to split the random matrix theory and the block diagonal inner product layer work into two separate papers. We have decided to submit only the block diagonal inner product layer work for this review.
>
> We plan to run experiments on the ImageNet dataset using AlexNet to demonstrate our work in a ‘large’ setting. However, we are currently waiting on a larger memory allocation on the Bridges supercomputer to handle this task. This allocation is granted, but will take up to 5 days to be active. We will update the paper with these results when we have them.
>
> Re: “[B]lock diagonal layers lead to more efficient networks. This point is murkier, because the paper doesn't discuss possible increases in *training time*.”
>
> All MNIST Lenet-5 experiments were run over 10000 iterations and all Cifar10 experiments were run over 9000 iterations. When implementing block diagonal inner product layers using method 1 without pruning we discussed the speed up of the weight matrix products for various matrix sizes. When implementing block diagonal inner product layers using method 2 with pruning, the speedup depends on the number of iterations it takes to fully prune to acheive the block structure. The pruning process itself only adds O(n/b) work to a layer with n weight parameters and b blocks in one iteration. We left the number of pruning iterations open as a hyperparameter.
>
> Re: “I had always thought pruning was done to decrease over-fitting, not to [decrease] computation speed, so this was a surprise to me; also note that the sparse matrix format can increase runtime if implemented as a sparse object, as demonstrated in this paper, but one could always pretend it is sparse, so you never ought to be slower with a sparse matrix”
>
> My understanding is that there are two primary reasons for pruning: to reduce overfitting and to reduce memory requirements. Reducing memory requiremens makes storing networks on mobile devices more feasible, for example. An unfortunate side effect is that sparse formats can greatly slow down computation speed.
>
> Re: “The authors seem to restrict themselves to convolutional networks in the first paragraph (and experiments) but don't discuss the implications or reasons of this assumption.”
>
> The focus on convolutional neural networks is simply because this is where the interest is. CNN’s are more powerful, successful so to convince readers I have focused on this kind of network. In 4.1 MNIST, Paragraph 4 I do touch on networks with only fully connected layers.
>
> Re: “I was also wondering about when 2 or more layers are block sparse, do these blocks overlap? i.e., are they randomly permuted between layers so that the blocks mix? And even with a single block, does it matter what permutation you use? (or perhaps does it not matter due to the convolutional structure?)”
>
> One can implement consecutive block layers. In the newest version of our paper we discuss a few ways to address the flow of information when consecutive layers are block. A general discussion can be seen in Section 4: Experiments, Paragraph 3. Comparisions are discussed in MNIST and Cifar10 experiment sections.
>
> Re: “Minor comment: last paragraph of 4.1, comparing with Sindhwani et al., was confusing to me. Why was this mentioned? And it doesn't seem to be comparable. I have no idea what "Toeplitz (3)" is.”
>
> We mention this because we deemed this work similar to our own and in need of comparison. We first mentioned Sindhwani et al. in Section 2: Related Work, Paragraph 3.
>
> We have uploaded a revised (working) version of the paper focusing on the block diagonal inner product layer results.
>
> We list the major changes here:
>
> Section 2: Related Work, Paragraph 3
> Here we focus on the related work we deem most similar to our own. We discuss Group Lasso, Group Convolution and Sindhwani et al. (2015) work with Toeplitz-like transforms as successful options to reduce network size in a structured manner.
>
> Section 4: Experiments, Paragraph 2
> We discuss the Figure 1. This new figure shows how block diagonal inner product layers scale for large weight matrices matrices.
>
> Section 4: Experiments, Paragraph 3
> Here we examine methods to improve the flow of information in sparse architecture and compare them to block diagonal inner product layer method 2 with pruning. These include fixed sub-block shuffling inspired by channel shuffling in Zhang et al. (2017) and random block shuffling.
>
> 4.2 CIFAR10, Paragraph 6
> Here we show that block diagonal inner product layer method 2 with pruning shows improved results over fixed sub-block shuffling inspired by channel shuffling in Zhang et al. (2017) and random block shuffling, which are other attempts to improve the flow of information in sparse architecture as discussed in Section 4: Experiments, Paragraph 3.

---

> > ### Comment · AnonReviewer2 · 2018-01-12
> > **thanks for your response**
> >
> > Thanks to the authors for responding (we have read it and taken it into account).

---

### Official Review · AnonReviewer1 · 2017-11-27
**A lack of synthesis.**

**Rating:** 4
**Confidence:** 3

**Review:**

This paper proposes replacing fully connected layers with block-diagonal fully connected layers and proposes two methods for doing so.  It also make some connections to random matrix theory.

The parameter pruning angle in this paper is fairly weak.  The networks it is demonstrated on are not particularly large (largeness usually being the motivation for pruning) and the need for making them smaller is not well motivated.  Additionally MNIST is a uniquely bad dataset for evaluating pruning methods, since they tend to work uncharacteristically well on MNIST (This can be seen in some of the references the paper cites).

The random matrix theory part of this paper is intriguing, but left me wondering "and then what?"  It is presented as a collection of observations with no synthesis or context for why they are important.  I'm usually quite happy to see connections being made to other fields, but it is not clear at all how this particular connection is more than a curiosity.  This paper would be much stronger if it offered some way to exploit this connection.

There are two half-papers here, one on parameter pruning and one on applying insights from random matrix theory to neural networks, but I don't see a strong connection between them. Moreover, they are both missing their other half where the technique or insight they propose is exploited to achieve something.

---

> ### Author Response · Authors · 2017-12-20
> **Split the paper into two separate papers**
>
> Thank you for your thorough review.
>
> We agree with your comment that the random matrix theory and the block diagonal inner product layer should be split into two separate papers.  We have split the paper along that line and decided to submit only the block diagonal inner product layer work for this review.
>
> Re: “The networks it is demonstrated on are not particularly large (largeness usually being the motivation for pruning) and the need for making them smaller is not well motivated.”
>
> We plan to run experiments on the ImageNet dataset using AlexNet to demonstrate our work in a ‘large’ setting. However, we are currently waiting on a larger memory allocation on the Bridges supercomputer to handle this task. This allocation is granted, but will take up to 5 days to be active. We will update the paper with these results when we have them.
>
> Re: “The parameter pruning angle in this paper is fairly weak.”
>
> Method 2 with pruning did perform better  in our experiments on Cifar10.  We also added some comparison to other methods that address limited information flow in sparse architecture.  When Bridges allows us to run experiments on Imagenet we will be sure to focus on the differences between method 1 and method 2.
>
> Re: “[This is] missing [the] other half where the technique or insight they propose is exploited to achieve something. ”
>
> We believe that using block diagonal layers in place of fully connected layers acheives a lot.  With block diagonal implementation, larger architectures are possible on hardware with memory constraints. We have condensed the storage requirements of a network without sacrificing execution time.
>
>
>
>
>
> We have uploaded a revised (working) version of the paper focusing on the block diagonal inner product layer results.
>
> We list the major changes here:
>
> Section 2: Related Work, Paragraph 3
> Here we focus on the related work we deem most similar to our own. We discuss Group Lasso, Group Convolution and Sindhwani et al. (2015) work with Toeplitz-like transforms as successful options to reduce network size in a structured manner.
>
> Section 4: Experiments, Paragraph 2
> We discuss the Figure 1.  This new figure shows how block diagonal inner product layers scale for large weight matrices matrices.
>
> Section 4: Experiments, Paragraph 3
> Here we examine methods to improve the flow of information in sprase architecture and compare them to block diagonal inner product layer method 2 with pruning. These inclue fixed sub-block shuffling inspired by channel shuffling in Zhang et al. (2017) and random block shuffling.
>
> 4.2 CIFAR10, Paragraph 6
> Here we show that block diagonal inner product layer method 2 with pruning shows improved results over fixed sub-block shuffling inspired by channel shuffling in Zhang et al. (2017) and random block shuffling, which are other attempts to improve the flow of information in sprase architecture as discussed in Section 4: Experiments, Paragraph 3.

---

### Public Comment · (anonymous) · 2017-11-17
**Related work**

The block diagonal inner product layer is rather similar with the group convolution in recent CNN architectures like ShuffleNet/Xception/MobileNet/ResNeXt/Inception... . In my understanding, a group convolution is just to make the parameter matrix of each filter from a dense matrix to a block diagonal one. They share similar advantages like the speedup and memory you mentioned in your paper. I think it would be better to have a discussion in your paper with these works.

---

> ### Author Response · Authors · 2017-11-17
> **Related work**
>
> We greatly appreciate your comment.
>
> We agree that group convolution should be mentioned in the final version and we are happy to make this change. It is an important, related idea that supports our work. However our understanding of group convolution is that the number of nonzero weights does not change, but rather it is the number of connections that changes. A particular set of filter weights does not see the output of every channel.
>
> Decoupling channels in images is natural because channels often carry very similar information.  When converting a fully connected layer to a block diagonal inner product layer, blocks may not even see a whole channel as is the case with 100 ip1 blocks on the MNIST dataset using the lenet-5 framework with 50 filters in conv2.

---

### Decision · Program_Chairs · 2018-01-29
**ICLR 2018 Conference Acceptance Decision**

**Decision:**

Reject

**Comment:**

The authors propose a technique for weight pruning that leaves block diagonal weights, instead of unstructured sparse weights, leading to faster inference.  However, the experiments demonstrating the quality of the pruned models are insufficient.   The authors also discuss connections to random matrix theory; but these connections are not worked out in detail.